# Cardiac MR segmentation based on sequence propagation by deep learning

**Chao Luo[1]◉, Canghong Shi[2]◉, Xiaoji Li[1]◉, Dongrui Gao[1]◉ ***

**1** Chengdu University of Information Technology, Chengdu, Sichuan, China, **2** School of Information Science and Technology Southwest Jiaotong University, Chengdu, Sichuan, China

◉ These authors contributed equally to this work.
* gdr1987@126.com

**Data Availability Statement:** All datasets files are available from the ACDC Challenge database (https://www.creatis.insa-lyon.fr/Challenge/acdc/databases.html). The authors did not receive special access privileges to the data. Heart Data Set of West China Hospital of Sichuan University, is

## Abstract

Accurate segmentation of myocardial in cardiac MRI (magnetic resonance image) is key to effective rapid diagnosis and quantitative pathology analysis. However, a low-quality CMR (cardiac magnetic resonance) image with a large amount of noise makes it extremely difficult to accurately and quickly manually segment the myocardial. In this paper, we propose a method for CMR segmentation based on U-Net and combined with image sequence information. The method can effectively segment from the top slice to the bottom slice of the CMR. During training, each input slice depends on the slice below it. In other words, the predicted segmentation result depends on the existing segmentation label of the previous slice. 3D sequence information is fully utilized. Our method was validated on the ACDC dataset, which included CMR images of *100* patients (*1700* 2D MRI). Experimental results show that our method can segment the myocardial quickly and efficiently and is better than the current state-of-the-art methods. When evaluating *340* CMR image, our model yielded an average dice score of *85.02 ± 0.15*, which is much higher than the existing classical segmentation method(Unet, Dice score = *0.78 ± 0.3*).

## Introduction

Heart disease has seriously threatened human health and is one of the diseases with the highest mortality rate [1] [2]. Accurate and rapid diagnosis of heart disease is very important to save lives. Cardiac Magnetic Resonance (CMR) has been widely used in the diagnosis and treatment of heart disease [3] [4]. However, manual segmentation and diagnosis are challenging due to factors such as low resolution of CMR and large interference noise of different tissue and organ parts [5]. Therefore, an automatic and accurate cardiac MRI segmentation method is highly desirable.

In recent years, a large number of methods based on deep learning have been widely used in medical image segmentation. This approach is reflected in the 2017 Automated Cardiac Diagnosis Challenge (ACDC) where the aim is to automatically perform segmentation and diagnosis on a 4D cine-CMR scan [6] [7]. In the challenge, all participants except the one participated in the deep learning method. Based on U-Net, FCN's classic method of segmentation

available from figshare: ([https://figshare.com/s/70021f845c4dc2a6e86d](https://figshare.com/s/70021f845c4dc2a6e86d)).

**Funding:** This work was supported by the National Natural Science Foundation of China (61602066), the Project of Sichuan Outstanding Young Scientific and Technological Talents (19JCQN0003), the major Project of Education Department in Sichuan (17ZA0063 and 2017JQ0030), and in part by the Natural Science Foundation for Young Scientists of CUIT (J201704) and the Sichuan Science and Technology Program (2019JDRC0077).

**Competing interests:** The authors have declared that no competing interests exist.

network was deeply explored. However, the sequence characteristics of the data set were not used in all the methods of participating in the competition. All participants' methods are based on 2D, ignoring the 3D sequence features [8]. The 2D based method divides the CMR by each slice, and the 3D convolution based method divides the CMR as a volume. The main reason why the 2D method is popular is that it does not require an oversized data set and is very light-weight. However, the 2D method does not make full use of the 3D sequence information. The 2D method treats each slice of the CMR as a separate image and inputs it into the network, thus ignoring the 3D sequence information between the slices [9].

In addition, the performance and robustness of 3D methods are poor. Through many experiments, ACDC participants have found that the performance of 2D method is always better than that of 3D method. In fact, the 3D method has many disadvantages: 1) The 3D method will result in a reduction in the number of data sets. The 3D method is to input the CMR as an image into the network, and the 2D method is input to the network according to each slice, which increases the number of images, thereby improving the performance and generalization of the network. 2) The implementation of the 3D method relies on 3D convolution. The boundary effect of 3D convolution will cause loss of sequence information, resulting in degradation of network performance. 3) The number of parameters of the 3D method is very large, it takes a lot of GPU memory, and it takes a lot of time to wait for the training result [10].

Therefore, how to effectively combine the advantages of the 2D method and the 3D method has become our focus. One possible method is to construct a 3D convolution-based network and simultaneously input sequence information of each slice based on 2D. In Biffi et al.(2019) [11], the author proposes a network based on 3D convolution combined with 2D slice images. The input of the network is a 3D MRI, and the 2D slice sequence image corresponding to the image is input before the upsampling operation. The method can effectively combine the advantages of the 2D method and the 3D method, not only increases the number of data sets, improves the performance and generalization ability of the network, but also effectively combines the sequence information. Through several comparison experiments, the author finds that this method effectively improves the performance, robustness and generalization ability of the network. In addition, another feasible method is to construct a network based on 2D convolution and combine 3D sequence information at the same time. In Zheng et al.(2018) [12], the author proposes a method based on 2D U-Net that can make full use of the 3D sequence information, which inputs each slice image of the CMR and simultaneously samples the previous slice image of the image.

In this paper, we propose a segmentation method based on 2D convolution for cardiac MRI. Our approach has three main contributions:

1. We propose a heart image segmentation network based on 2D method, which can effectively utilize sequence information. Our method not only can effectively increase the number of data sets, but also greatly reduces the number of parameters of the network and reduces the training time.

2. Because our method increases the number of data sets, our method is more powerful than the 3D method, and has excellent robustness and generalization ability.

3. Since we have adopted a 2D-based convolution method, our method makes full use of the sequence information while lowering the computational resources compared with the 3D convolution method. Our proposed method effectively overcomes the shortcomings of the 3D method.

Our proposed method was validated on the ACDC dataset. The experimental results show that our method can segment the myocardial region of the heart quickly and accurately.

Compared with the classical segmentation method, our method has better performance, stronger robustness and generalization ability.

## Related work

Due to the special physiological structure of the heart, there are a large number of watery cysts in the heart disease area. The pixels in the cyst area will cause the image pixels in the lesion area to drop sharply, and the gray level difference from the normal area will decrease. Therefore, precise segmentation of the heart area is very challenging.

### Traditional methods

Traditional image segmentation methods mainly include threshold-based methods (histogram bimodal method, dynamic programming), pixel-based classification (clustering, Gaussian mixture model), edge-based method (Canny edge detection, Harris corner detection), based on Regional methods (watershed) and morphological-based image segmentation methods. The inherent drawbacks of the above methods themselves make it difficult to extract regions of interest directly from high noise, low contrast cardiac images. Zheng et al. proposed a machine learning-based segmentation method that extracts a set of geometric and image features and then uses a Probabilistic Enhancement Tree (PBT) to train the classifier for segmentation [13]. David et al. proposed a method based on the hidden semi-Markov model (HSMM) and support vector machine (SVM), which is used to segment the main heart sounds in the electrocardiogram (PCG) [14].

### Deep learning methods

As we all know, before *2013*, there was basically no deep learning method to analyze cardiac MRI. However, with the development of deep learning, in the second kaggle challenge in 2015, a large number of deep learning methods for cardiac MRI processing appeared. Most papers are based on 2D Convolutional Neural Networks (CNN) to analyze MRI data. For exmple, Duan et al. proposed a 2D-based full convolutional network [15]. This approach combines the ability to resolve 3D spatial consistency. Moreover, a refinement step is designed to explicitly impose shape prior knowledge and improve segmentation quality. This step is effective for overcoming image artifacts (e.g., due to different breath-hold positions and large slice thickness), which preclude the creation of anatomically meaningful 3D cardiac shapes. Therefore, it can effectively segment the myocardial region. In addition, Emad et al. proposed a path-based CNN to segment the CMR of the left ventricle [16]. Kong et al. proposed a method of combining 2D CNN with recurrent neural network (RNN) to identify end-diastolic and end-systolic phases [17]. Bai et al. proposed a segmentation method based on the full convolutional neural network (FCN), which is trained and verified on the largest CMR dataset, and the experimental results can reach the level of human experts [18]. Baumgartner et al. propose a method based on 2D convolutional networks and a method based on 3D convolutional networks that provided good results on the ACDC 2017 [19].

## Model and loss

Our method mainly consists of two modules: context extraction module and segmentation module. The first part mainly extracts the context features of the image, and depends on the sequence information of the data set. The second part is the segmentation module. Inspired by U-Net, we designed a segmentation network based on the Encoder-Decoder structure. In addition, the loss function we use will be introduced in this section.

## Model architecture

As shown in Fig 1, our proposed network consists of two modules: context feature extraction module and a segmentation module. As shown on the left side of Fig 1, the context feature extraction module can efficiently extract the context features implied between images.

**Context feature extraction module.** There are *3* paths in the input of the context: Label[i-1] is the ground-truth of the previous slice of the input image, Ori[i-1] is the original image of the previous slice of the input image, Ori[i+1] is input an image of the next slice of the image. If Ori[i] is the first slice image in the image sequence, then Ori[i-1] is empty, and Ori[i-1] is set to null. If Ori[i] is the last slice image in the image sequence, then Ori[i+1] does not exist, and Ori[i+1] is set to null.

The main structure of the module includes convolution, BN, RELU and deep supervision. We learn the characteristics of different fields of view of the image by designing convolutions with different convolution kernel sizes. And by designing with different numbers of kernels, the contextual features of the image can be fully learned. Improve the utilization of image data and avoid feature omissions. Using the BN layer after each convolutional layer can effectively improve the network fitting speed, improve the stability of the network during the training process, and solve the gradient dispersion problem of the network in the back propagation process. RELU can improve the nonlinearity of the network and enhance the generalization ability of the network. The pooling layer can limit the memory requirements and improve the training speed of the network. In addition, the features between the layers corresponding to the three paths are merged with each other, which is more conducive to the learning of the context features between the images, and effectively avoids feature loss.

**Segmentation module.** Inspired by U-Net, we designed a network for image segmentation. The network contains two paths: Encoder path and Eecoder path. As shown in the right

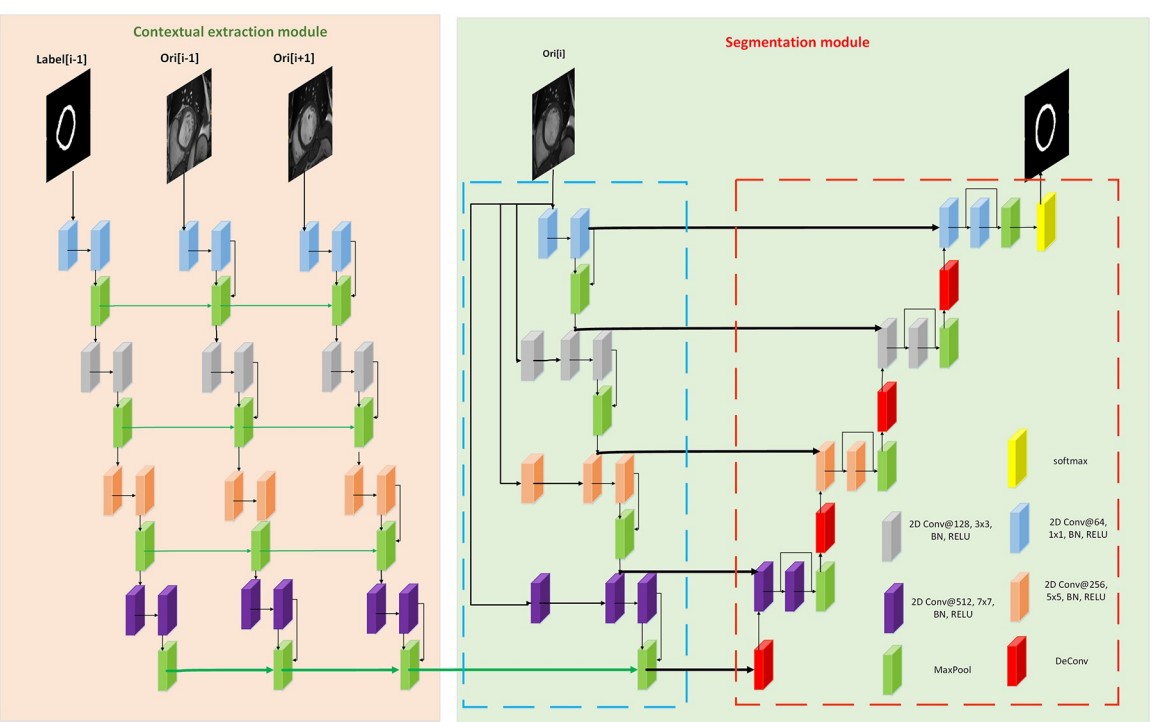

**Fig 1. Our network consists of context feature extraction module and segmentation module.**

part of Fig 3, the blue dotted box is the Encoder path. The path consists of four convolutional layers of different convolution kernel sizes, each containing convolution, BN, RELU and POOL. And each layer incorporates the original image features from the first layer, which effectively avoids feature loss. The red dotted box is the Decoder path. The path consists of four upsampling layers, each of which contains one deconvolution layer, two convolutional layers, and one pooling layer. Each of the upsampling layers is fused with the features of the corresponding convolution layer in the Encoder path. The Decoder path can restore the feature size to the same size as the input image, enabling the network to perform end-to-end training, that is, inputting an image to be segmented to the network and directly output a segmented image that is the same size as the original image.

## Loss function

The cross entropy function is mainly used to classify and segment tasks. Herein, we make use of the cross entropy as the loss function of the myocardial segmentation. Formally, it is defined as:

$$Loss_{CE} = -\frac{1}{m}\sum_{i}^{m}[y^i \log p^i + (1 - p^i) \log(1 - y^i)] \tag{1}$$

where $m$ is the number of samples, $y^i$ is the label of the sample, and $p^i$ is the predicted probability value, $p_i \in (0, 1)$. Furthermore, the cross entropy loss function is put to a frequent use in classification networks. In the current paper, we employ the classification methodology for the purpose of segmenting tasks. We take into consideration the myocardial circle to be segmented as a category and the background as a category.

## Experiments and discussion

### Dataset and preprocessing

In this experiment, we used the data set published in the ACDC competition. To validate our method, dice similarity coefficien(DSC), area under the curve(AUC), jaccard similarity coefficient(JSC) and F1 score are used to measure the results of the segmentation.

**Dataset.** In the experiment of this paper, we used the myocardial data set of the ACDC competition(left ventricle, myocardium and right ventricle). The ACDC dataset was created from actual clinical examinations obtained at Dijon University Hospital, which covers several well-defined pathologies with sufficient cases:1) Proper training in machine learning methods. 2) Clearly assess changes in the main physiological parameters obtained from cine-MRI(in particular diastolic volume and ejection fraction). The corresponding database is composed by *100* patients with 3D cine-MR datasets acquired in clinical routine.

**Preprocessing.** Since the ACDC dataset is a 3D MRI, in order to increase the number of datasets, we made each slice of the dataset into a single 2D image with a total of *1,700* 2D images. The spatial resolution of each CMR image is *1.37 × 1.68* mm and the size is between *222 × 224* and *216 × 256*. Considering that the myocardium is small and there is a lot of noise around, we first resample the resolution of the image to *1 × 1* mm, and then process the size of each image to a fixed size of *128 × 128*.

### Evaluation metrics

We use of four indicators for the purpose of measure the performance of the network, which includes the dice similarity coefficient (DSC), area under the curve (AUC), Jaccard similarity coefficient (JSC), and F1-score for the assessment of the segmentation accuracy. The DSC was mostly employed for the calculation of the overlap metric between the results of segmentation

and the ground truth. The DSC for bit vectors was defined as:

$$DSC = \frac{2 \parallel PG \parallel_2}{\parallel P \parallel_2 + \parallel G \parallel_2} \tag{2}$$

where $PG$ is the element-wise product of the prediction ($P$) and the ground truth ($G$), and $\parallel x \parallel_2$ is the L2-norm of $x$. The AUC is a probability value. The greater the AUC value, the better the performance. The AUC score was computed with a closed-form formula:

$$AUC = \frac{S_0 - n_0(n_0 + 1)/2}{n_0 n_1} \tag{3}$$

where $n_0$ is the number of pixel that belong to the ground truth, $n_1$ is the opposite and $S_0 = \sum_{i=1}^{n_0 r_i}$, where $r_i$ is the rank given by the predict model of the ground truth to the $i$th pixel in the CMR image.

The F1-score is the harmonic average of precision and recall, wherein an F1-score reaches its best value at one (perfect precision and recall) and the worst at zero.

$$F1 - score = \frac{2 \times Precisition \times Recall}{Precision + Recall} \tag{4}$$

The JSC is put to use for the improvement of similarities and differences between finite sample sets. The larger the JSC value, the higher the sample similarity.

$$JSC = \frac{|P \cap G|}{|P| + |G| - |P \cap G|} \tag{5}$$

where $P$ is the probability of prediction, $G$ is the ground-truth.

## Implementation details and parameter settings

In order to ensure the stability and efficiency of the experiment, we conducted several experiments to explore the optimal settings of the parameters. Finally, the optimal parameter configuration scheme we adopted is as follows.

**Train strategy.**   To facilitate training, we use a 5-fold cross-validation method to divide the data set. Among them, *1020* MRIs were used as training sets, *340* MRIs were used as test sets, and *340* were used as verification sets. The validation set is not used for training and testing and is only used to finally verify the performance of the model.

**Learning rate strategy.**   We conducted several trials with different learning rates, and the results showed that the learning rate of *0.001* was the most appropriate. Therefore we set the learning rate to *0.001*, the initial learning rate is exponentially degraded every 10 iterations at a learning rate decay rate of 0.9.

**Experiment configurations.**   To ensure the consistency of the experiment, we use accuracy as the quantization metric, *100* epochs are trained for each experiment (*batch_size* = 5). All experiments are implemented in python2.7 by using Tensorflow and Keras framework. We train the networks on a NVIDIA Tesla M40 GPU and the model that performs the best on validation data set are saved for further analysis.

## Result

To verify the performance of our proposed network, we compared it to three classic segmentation methods (U-NET, Deeplabv3, and SegNet). Table 1 shows the segmentation results for each network. As can be seen from the table below, our proposed method shows the best

**Table 1. Experimental results of 5 different networks.** The table shows the average of the four indicators. Ours-no-cem means to remove the context extraction module in our proposed method.

| Model | DSC | *DSC(std)* | AUC | F1-score | JSC |
|---|---|---|---|---|---|
| SegNet | 0.7211 | 0.08 | 0.7509 | 0.7673 | 0.5693 |
| Deeplabv3 | 0.7563 | 0.13 | 0.8246 | 0.7315 | 0.6383 |
| U-Net | 0.7806 | 0.06 | 0.8845 | 0.7821 | 0.6792 |
| Ours-no-cem | 0.8009 | 0.05 | 0.8513 | 0.8455 | 0.7738 |
| **Ours** | 0.8768 | 0.02 | 0.9330 | 0.8791 | 0.7924 |

performance. Based on the five-fold cross-validation training strategy, our method has an average DSC value of *0.8768*, an average AUC value of *0.9330*, an average F1-score of *0.8791*, and an average JSC of *0.7924*. The values of these four indicators are much higher than the other three classic segmentation methods. Compared with U-Net, the DSC value of our method is much higher than U-Net's *0.7806*, an increase of *7%*. And our method's AUC value, F1-score and JSC values are much higher than U-Net. In addition, from the comparison of the data in Table 1, it can be clearly seen that our method is much higher than DeeplabV3 and SegNet in these four indicators. Therefore, compared to U-Net, Deeplabv3 and SegNet, our method performance is far superior to the other three methods. In addition, the dice value of the context extraction module removed from our method is much lower than the dice value of this method.

Fig 2 shows a block diagram of the DSC values for the four networks. As can be seen from the figure, the span of ours method's block diagram is small, and the DSC values of the five

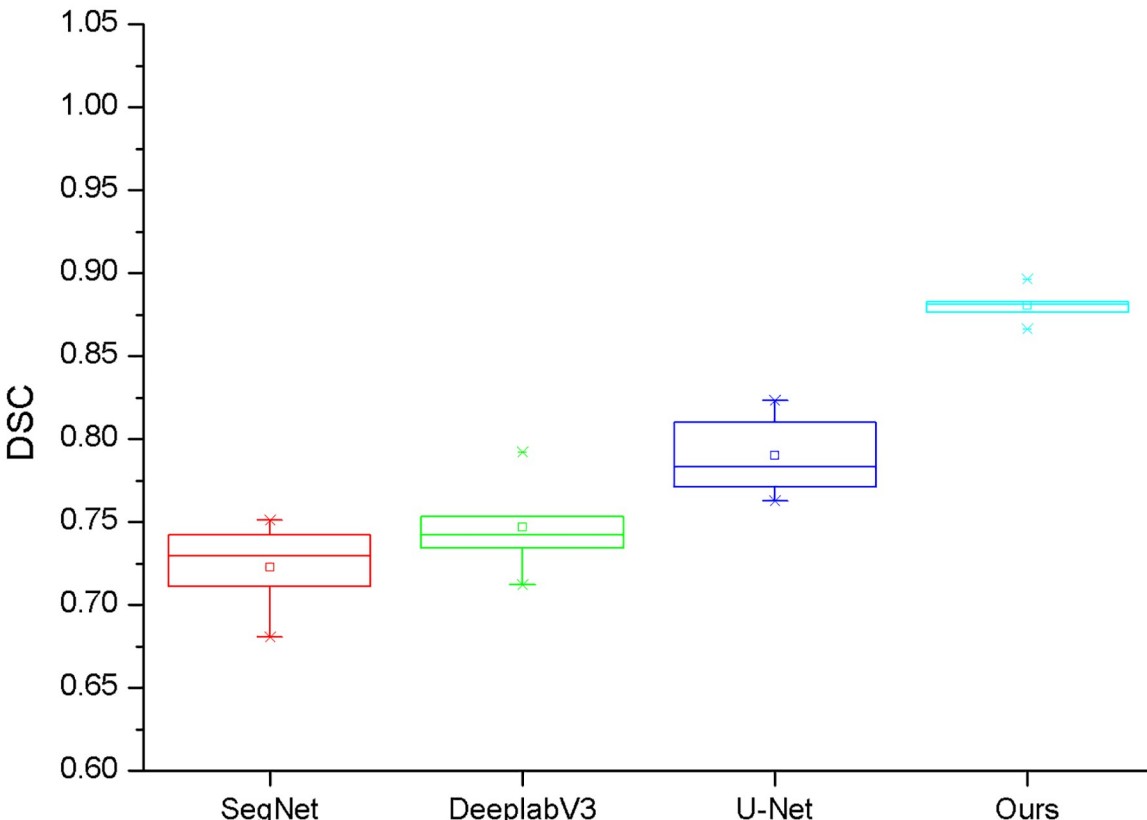

**Fig 2. Box diagram of DSC for four different networks.** Each network performs 5 experiments, and the box plot was drawn based on the DSC value of each experimental result.

experiments are mainly concentrated around *0.87*. However, the span of U-Net, SegNet and Deeplabv3 is much larger than our approach. Therefore, it is clear that our method not only performs better than the other three methods but is more stable and robust.

Fig 3 show dice indicator curve changes for four different methods. In this experiment, experiments were carried out on four networks using a 5-fold cross-validation experimental method, and the results of each experiment were recorded and plotted. As shown in Fig 3, the DSC value of each of our methods is higher than that of the other three networks, and the fluctuation range is the smallest. It can be seen that our method not only has good segmentation performance, but also has better stability and robustness.

Based on the comparison of the dice indicators and analysis, we propose that the proposed method has good myocardial segmentation performance and can accurately and quickly segment the muscle region of the heart. Compared with the three classic segmentation methods of DeeplabV3, U-Net and SegNet, our method has far more performance than the three methods in myocardial segmentation tasks, and has good stability and robustness.

It can be clearly seen from Fig 4 that our method can effectively and accurately segment the myocardial region. In the first row comparison chart in Fig 4, compared with the manually label, SegNet segmentation effect is very poor, *1/3* of the region is not correctly segmented, DeeplabV3 does not completely segment the myocardial region, U-Net There is also no complete correct segmentation of the myocardium, and our method is able to accurately and

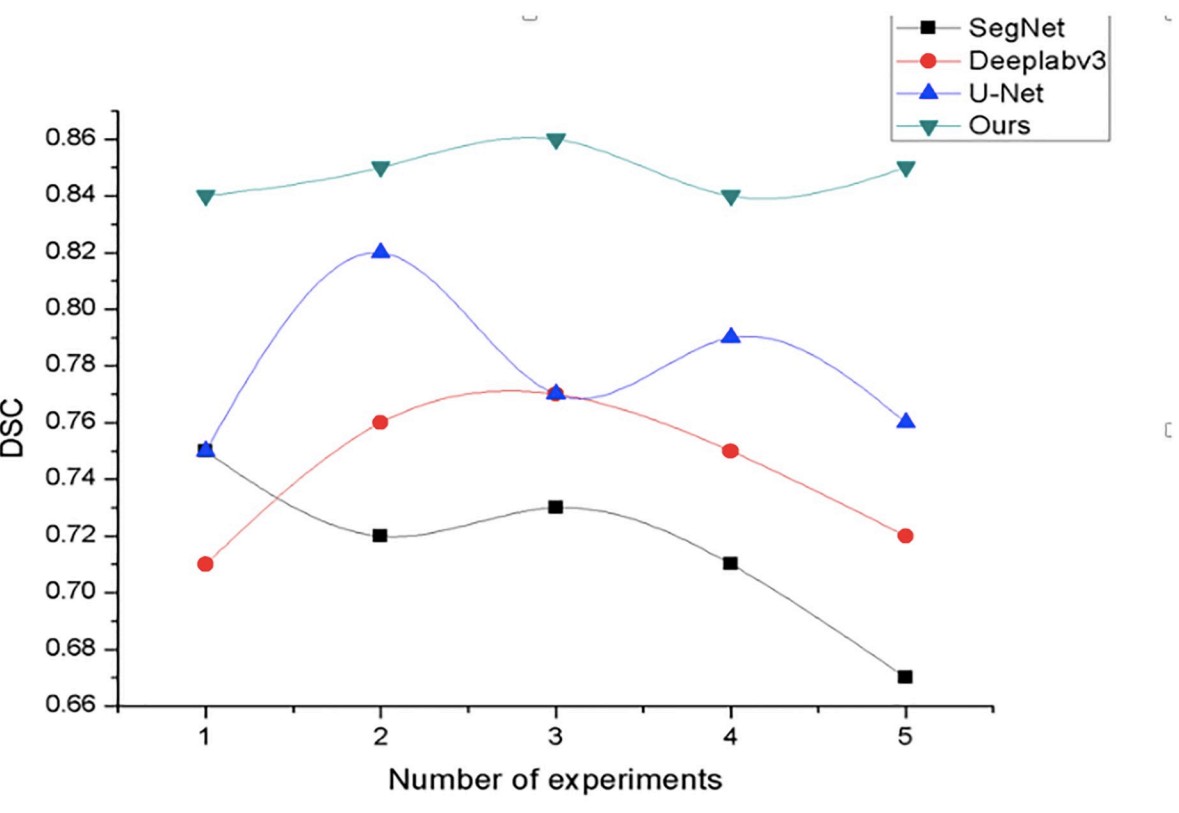

**Fig 3. DSC curve change chart.**

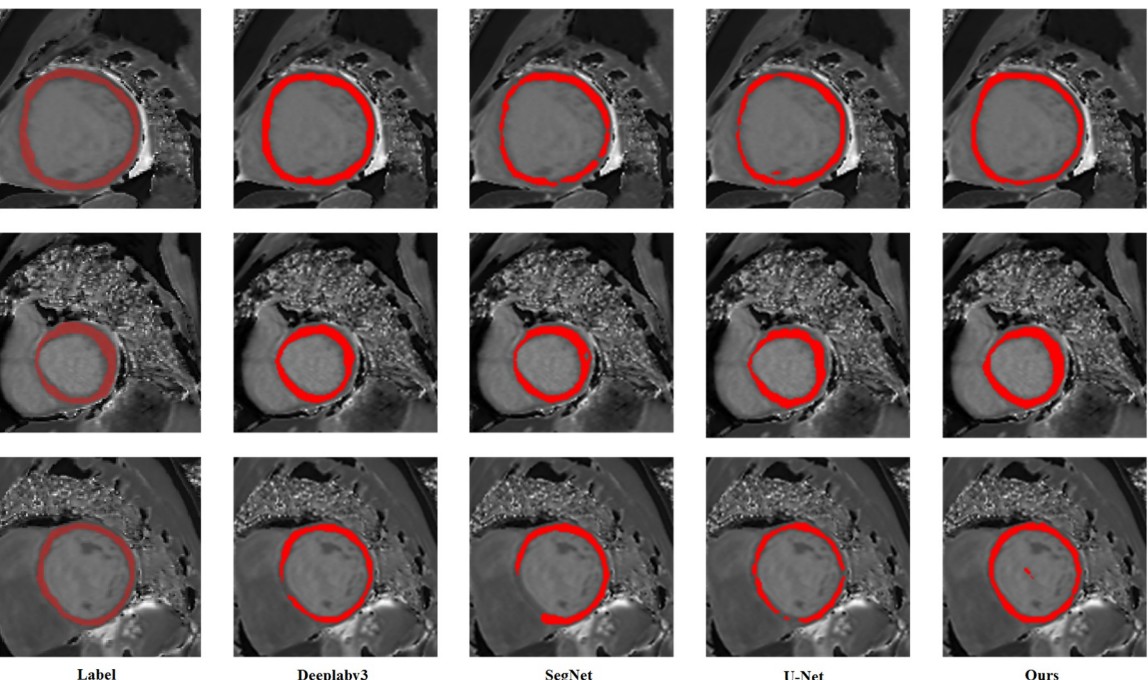

**Fig 4. Segmentation results of three different samples in different networks.**

completely segment the myocardium. Similarly, we can compare the results of the second and third row segmentation. SegNet, DeeplabV3 and U-Net can not accurately segment the myocardial region, and it will produce interference noise, which will affect the doctor's diagnosis. Our method is able to accurately and completely segment the smooth myocardium without disturbing noise, which is closer to the label image.

In addition, in order to verify the robustness of our proposed method, we performed experiments on cardiac MRI images of *150* patients provided by West China Hospital of Sichuan University. We converted 3D MRI from *150* patients to *1350* available 2D MRI. The experiment was completed using a 5-fold cross-validation data partitioning strategy. Among them, *810* cases served as the training set, *270* cases served as the test set, and 270 cases served as the verification set. Fig 5 shows our results. From the Fig 5, our method can also accurately and effectively segment the myocardial region.

Table 2 shows the results of our method and comparison method in the data set of West China Hospital of Sichuan University. From the table, we can know that our method is higher than the comparison method in all indicators.

In addition, to further verify the robustness of our proposed method. We performed experiments on the left ventricle dataset and the right ventricle dataset (ACDC). From the results in Table 3, we can know that our proposed method has excellent performance on the left ventricular data set, and the four indicators are much higher than the other four comparison methods. From Fig 6, we can visually see that the segmentation effect of our method is better.

Table 4 is the result of right ventricle segmentation of our method. From the comparison of the indicators in the table, we can know that the dice value of our method in the right ventricle segmentation is 0.93, which is far more than the other three segmentation methods. Fig 7 is an

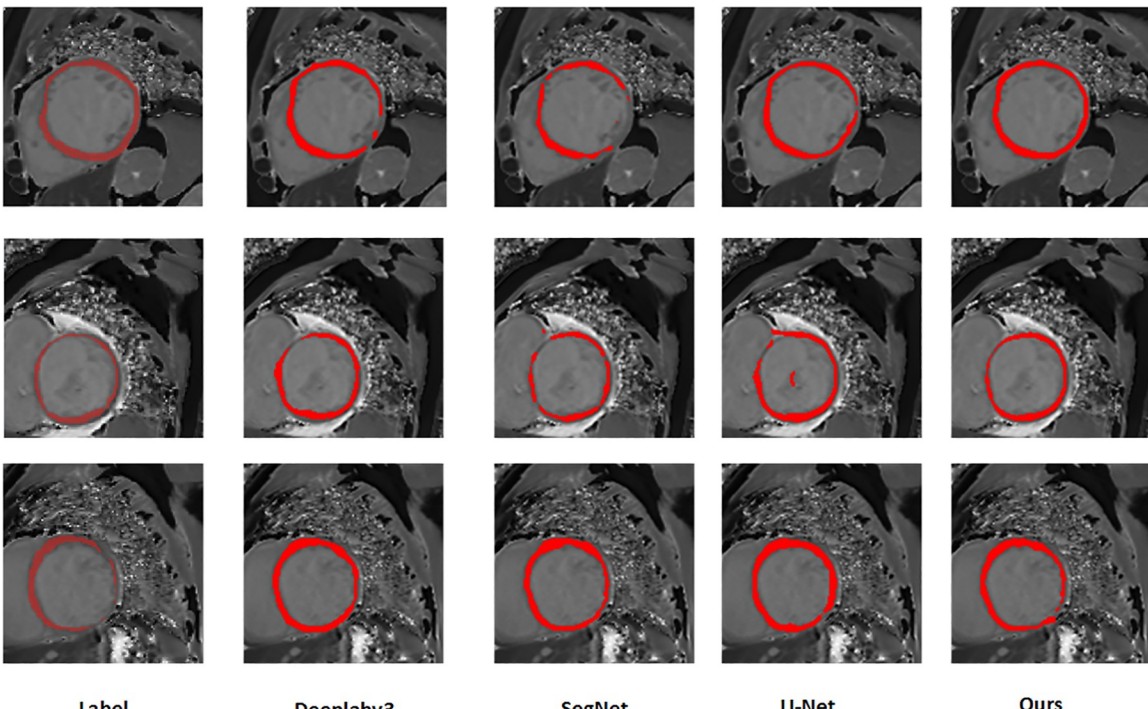

**Fig 5. Segmentation results of three different samples in different networks with the data of West China Hospital of Sichuan University.**

**Table 2. The experimental results in the data set of West China Hospital of Sichuan University.**

| Model | DSC | DSC(std) | AUC | F1-score | JSC |
|---|---|---|---|---|---|
| SegNet | 0.7362 | 0.07 | 0.78233 | 0.8321 | 0.6315 |
| Deeplabv3 | 0.7746 | 0.08 | 0.8560 | 0.7613 | 0.6859 |
| U-Net | 0.8034 | 0.04 | 0.8742 | 0.8124 | 0.7386 |
| Ours | 0.8923 | 0.02 | 0.9510 | 0.93145 | 0.8378 |

**Table 3. Index results of segmentation in the left ventricle.**

| Model | DSC | DSC(std) | AUC | F1-score | JSC |
|---|---|---|---|---|---|
| SegNet | 0.7585 | 0.07 | 0.8053 | 0.84867 | 0.6834 |
| Deeplabv3 | 0.7865 | 0.09 | 0.8368 | 0.7062 | 0.7112 |
| U-Net | 0.8335 | 0.05 | 0.8246 | 0.8523 | 0.7709 |
| Ours | 0.9035 | 0.03 | 0.9123 | 0.8911 | 0.8463 |

example of right ventricle segmentation. We can visually see that the right ventricle segmented by our method is closer to the label and smoother.

## Conclusion

In this paper, we propose a network for image segmentation with sequence propagation features. The method mainly includes two modules: contextual extraction module and

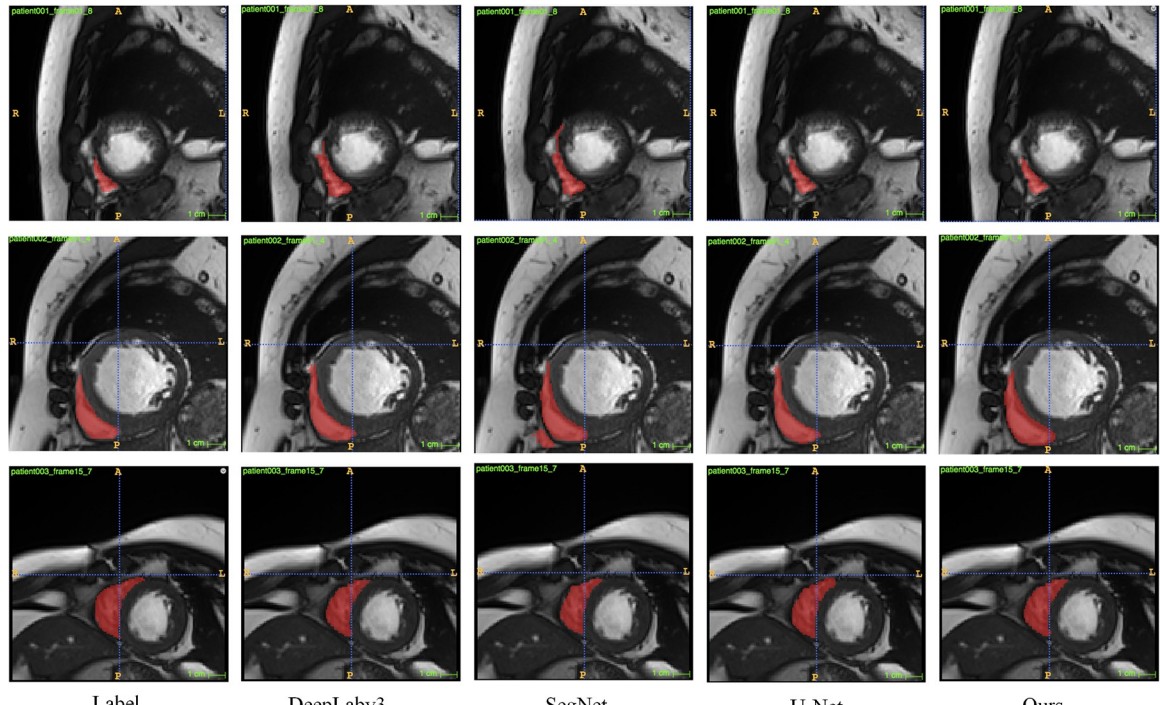

| Label | DeepLabv3 | SegNet | U-Net | Ours |

**Fig 6. Segmentation effect in the left ventricle.**

segmentation module. The context extraction module can fully extract the context features of the image to be segmented, and effectively combines the sequence features. The segmentation module is an encoder-decoder module, and inputting an image can directly predict a segmented image. The module effectively learns the characteristics of the original image and avoids feature loss and gradient dispersion by the design of the jump connection. To prove the validity of our proposed method, we compared it to SegNet, DeeplabV3 and U-Net. The experimental results show that our network can accurately and quickly segment the myocardial region of the heart. Compared with the other three classical segmentation networks, our network segmentation performance is better and more robust. The network is also suitable for the segmentation tasks of other medical images. In future research, we will use this network to accomplish different medical image segmentation tasks.

**Table 4. Index results of segmentation in the right ventricle.**

| Model | DSC | DSC(std) | AUC | F1-score | JSC |
|---|---|---|---|---|---|
| SegNet | 0.7761 | 0.05 | 0.8262 | 0.8522 | 0.7159 |
| Deeplabv3 | 0.8003 | 0.06 | 0.8491 | 0.7558 | 0.7624 |
| U-Net | 0.8522 | 0.03 | 0.8694 | 0.8722 | 0.8104 |
| Ours | 0.9353 | 0.01 | 0.8921 | 0.9134 | 0.8787 |

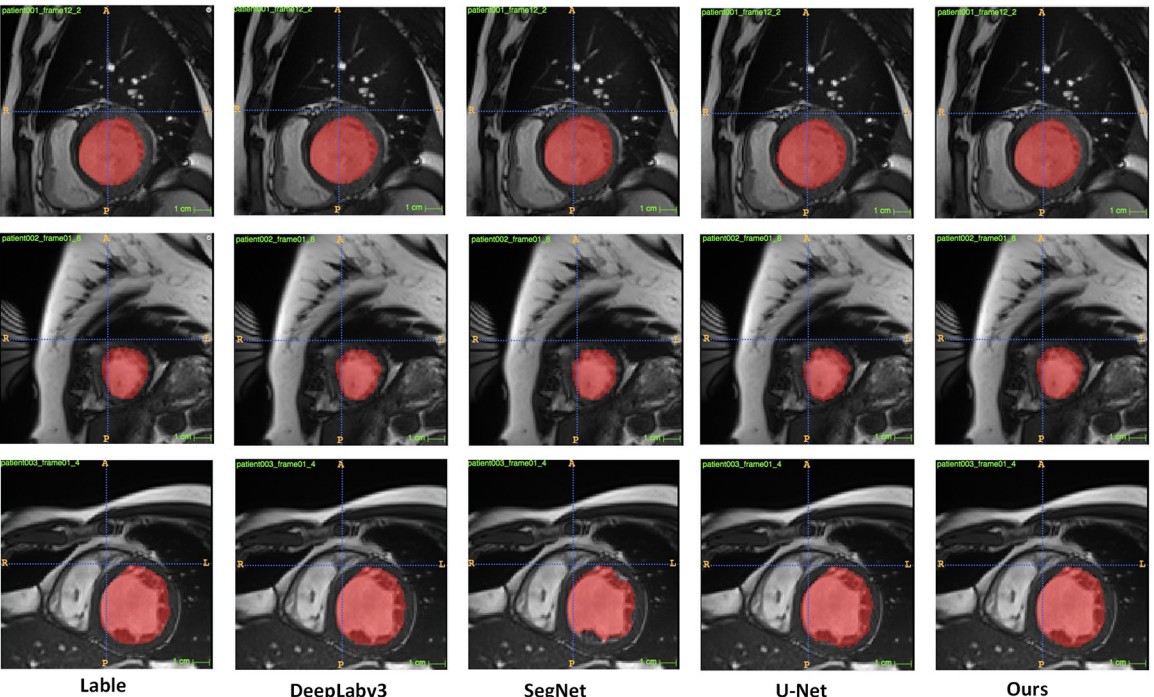

|           |           |        |       |      |
|-----------|-----------|--------|-------|------|
| **Lable** | **DeepLabv3** | **SegNet** | **U-Net** | **Ours** |

**Fig 7. Segmentation effect in the right ventricle.**

## Author Contributions

**Data curation:** Chao Luo, Canghong Shi, Xiaoji Li, Dongrui Gao.

**Formal analysis:** Xiaoji Li, Dongrui Gao.

**Funding acquisition:** Xiaoji Li, Dongrui Gao.

**Investigation:** Xiaoji Li, Dongrui Gao.

**Methodology:** Chao Luo, Canghong Shi, Xiaoji Li, Dongrui Gao.

**Project administration:** Dongrui Gao.

**Supervision:** Canghong Shi, Dongrui Gao.

**Validation:** Chao Luo, Xiaoji Li, Dongrui Gao.

**Visualization:** Canghong Shi, Dongrui Gao.

**Writing – original draft:** Chao Luo, Xiaoji Li, Dongrui Gao.

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
