## [Decision Letter · Decision Letter 0]

30 Aug 2019

PONE-D-19-18130

Cardiac MR Segmentation Based on Sequence Propagation by Deep Learning

PLOS ONE

Dear Dr. Gao,

Thank you for submitting your manuscript to PLOS ONE. After careful consideration, we feel that it has merit but does not fully meet PLOS ONE’s publication criteria as it currently stands. Therefore, we invite you to submit a revised version of the manuscript that addresses the points raised during the review process.  In particular, both reviewers raised major concerns about reviewing similar methods in the literature, and providing a more rigorous comparison of the proposed method to previous approaches.  In addition, the source code for the proposed algorithm should be made available on a public repository, in accordance with PLOS ONE's policy on Materials and Software Sharing. 

We would appreciate receiving your revised manuscript by Oct 14 2019 11:59PM. To enhance the reproducibility of your results, we recommend that if applicable you deposit your laboratory protocols in protocols.io, where a protocol can be assigned its own identifier (DOI) such that it can be cited independently in the future. For instructions see: http://journals.plos.org/plosone/s/submission-guidelines#loc-laboratory-protocols

We look forward to receiving your revised manuscript.

Kind regards,

Dzung Pham

Academic Editor

PLOS ONE

Journal Requirements:

Reviewers' comments:

Reviewer's Responses to Questions

**Comments to the Author**

1. Is the manuscript technically sound, and do the data support the conclusions?

Reviewer #1: Yes

Reviewer #2: Partly

2. Has the statistical analysis been performed appropriately and rigorously? 

Reviewer #1: No

Reviewer #2: No

3. Have the authors made all data underlying the findings in their manuscript fully available?

Reviewer #1: Yes

Reviewer #2: Yes

4. Is the manuscript presented in an intelligible fashion and written in standard English?

Reviewer #1: Yes

Reviewer #2: Yes

5. Review Comments to the Author

Reviewer #1: Chao et al proposed a sequence propagation-based U-net method for cardiac image segmentation. The method can effectively segment from the top slice to the bottom slice of the CMR Image. Each input slice depends on previous slice in the training process. Therefore, the predicted segmentation result will be conditioned on the existing segmentation labels, which effectively propagates adjacent information. The method is interesting and promising. However, there are problems in the paper.

1: The literature review is not thorough (IMPROTANT). There are lots of related work that have not been mentioned in this paper. For example, the following works were all focusing on cardiac segmentation.

Duan, J., Bello, G., Schlemper, J., Bai, W., Dawes, T.J., Biffi, C., de Marvao, A., Doumou, G., O’Regan, D.P. and Rueckert, D., 2019. Automatic 3D bi-ventricular segmentation of cardiac images by a shape-refined multi-task deep learning approach. IEEE T. Med. Imaging, 2019.

O. Bernard, A. Lalande, C. Zotti, et al., “Deep learning techniques for automatic mri cardiac multi-structures segmentation and diagnosis: Is the problem solved?,” IEEE T. Med. Imaging, 2018.

W. Bai, M. Sinclair, G. Tarroni, O. Oktay, M. Rajchl, G. Vaillant, A. Lee, N. Aung, E. Lukaschuk, M. Sanghvi, et al., “Human-level cmr image analysis with deep fully convolutional networks,” J. Cardiov. Magn. Reson., 2018.

C. Baumgartner, L. Koch, M. Pollefeys, and E. Konukoglu, “An exploration of 2d and 3d deep learning techniques for cardiac mr image segmentation,” ArXiv Preprint ArXiv:1709.04496, 2017.

J. Patravali, S. Jain, and S. Chilamkurthy, “2d-3d fully convolutional neural networks for cardiac mr segmentation,” ArXiv Preprint ArXiv:1707.09813, 2017.

H. Winther, C. Hundt, B. Schmidt, C. Czerner, J. Bauersachs, F. Wacker, and J. Vogel, “V-net: Deep learning for generalized biventricular cardiac mass and function parameters,” ArXiv Preprint ArXiv:1706.04397, 2017.

Duan, J., Schlemper, J., Bai, W., Dawes, T.J., Bello, G., Doumou, G., De Marvao, A., O’Regan, D.P. and Rueckert, D., 2018, September. Deep nested level sets: Fully automated segmentation of cardiac MR images in patients with pulmonary hypertension. In MICCAI (pp. 595-603). Springer, Cham, 2018.

2. Please discuss the difference in detail between your method and the one in reference [12]

3. In your first contribution, you mentioned segmentation of a heart, which includes RV, LV and Myocardium. However, in experiments, only myocardium is segmented, why? Can you method segment all anatomies? If so please conduct experiments.

4. Only one dataset (ACDC) was used. I would like to see experiments on different datasets, such as Sunnybrook cardiac data and 2011 LV segmentation challenge.

http://www.cardiacatlas.org/challenges/lv-segmentation-challenge/

http://www.cardiacatlas.org/studies/sunnybrook-cardiac-data/

5. How accurate is the method in terms of extracting clinical associated measures, such as stroke volume, and cardiac output, ejection fraction, etc. How about the segmentation accuracy for apical and basal slices. I would like to see both quantitative and quality results on these.

6. In experiments, I would like to see the quantitative measures based on Hausdorff distance and mean contour distance.

Reviewer #2: This works proposes new deep learning approach for Cardiac MR Segmentation based on the classical U-Net architecture (2D convolutions) where the sequence information is integrated. Also, the authors claims that the proposed method is better than the current state-of-the-art-methods. The proposed method is in general technically correct and seems to provide good results in the evaluated dataset. However, I have some concerns about the novelty of this approach. The most severe flaw of the manuscript is the lack of an 'honest' comparison with other segmentation strategies in the literature.

Additional comments:

1) The authors claims that the proposed method is better than the current state-of-the-art-methods. The authors are suggested to include a small revision of the literature and compare their results with the bibliography (include the number of patients used for validation for each method). For instance, in [1], [2] or [3] the authors achieve similar dice's values as the one proposed in this paper. Moreover, most of the methods described in [4] achieve a dice values for myocardial tissue classification greater than 0.85.

2) The architecture described in this work is in some way similar to others approaches proposed in the literature exactly in the same context as this paper. For example, in [3] the authors propose a way to introduce contextual information into the NN. And in [1] the authors studied the effects of introducing the original information into the encode path of the u-net. Can the authors discuss the differences between those approaches and the one proposed?

3) Did the authors consider using the Dice coefficient or Jaccard as the objective function, instead of the cross entropy ? And why?

4) Given that little training data is available, why are the authors not using data augmentation ?

5) The paper should describe precisely how the dataset were selected for training, validation and testing.

5.a) Were the dataset split by patients or slices ?

5.b) How many images were used for validation (i.e. training the hyperparameters) ?

5.c) Was the validation dataset the same as the one used for testing or training ?

6) Remove or translate the comment to english in Fig. 9

7) First sentence of the 4 paragraph (line 280) is not clear, it should be rewritten.

8) The authors are suggested to include a new plot with the dice accuracy for each epoch (mean value of the 5-fold cross validation over validation dataset) for the architectures studied.

Minors comments:

-) There are some typos, for example, missing dot in line 290.

-) The block diagram shows in Fig. 5 is commonly named boxplot.

-) For non-image content I strongly recommend to use vectorized formats such as pdf or eps.

[1] A. H. Curiale, F. D. Colavecchia, and G. Mato, “Automatic quantification of the lv function and mass: A deep learning approach for cardiovascular mri,” Computer Methods and Programs in Biomedicine, vol. 169, pp. 37 – 50, 2019.

[2] L. K. Tan, R. A. McLaughlin, E. Lim, Y. F. Abdul Aziz, and Y. M. Liew, “Fully automated segmentation of the left ventricle in cine cardiac mri using neural network regression,” Journal of Magnetic Resonance Imaging, vol. 48, no. 1, pp. 140–152, 2018.

[3] Q. Zheng, H. Delingette, N. Duchateau, and N. Ayache, “3d consistent & robust segmentation of cardiac images by deep learning with spatial propagation,” IEEE Transactions on Medical Imaging, 04 2018.

[4] O. Bernard, A. Lalande, C. Zotti, F. Cervenansky, X. Yang, P. Heng, I. Cetin, K. Lekadir, O. Camara, M. A. Gonzalez Ballester, G. Sanroma, S. Napel, S. Petersen, G. Tziritas, E. Grinias, M. Khened, V. A. Kollerathu, G. Krishnamurthi, M. Roh ´e, X. Pennec, M. Sermesant, F. Isensee, P. J ¨ager, K. H. Maier-Hein, P. M. Full, I. Wolf, S. Engelhardt, C. F. Baumgartner, L. M. Koch, J. M. Wolterink, I. Iˇsgum, Y. Jang, Y. Hong, J. Patravali, S. Jain, O. Humbert, and P. Jodoin, “Deep learning techniques for automatic mri cardiac multi-structures segmentation and diagnosis: Is the problem solved?,” IEEE Transactions on Medical Imaging, vol. 37, pp. 2514–2525, Nov. 2018.

6. PLOS authors have the option to publish the peer review history of their article (what does this mean?). If published, this will include your full peer review and any attached files.

Reviewer #1: Yes: Jinming Duan

Reviewer #2: No

---

## [Author Response · Author response to Decision Letter 0]

6 Oct 2019

Reviewer 1: 

1: The literature review is not thorough (IMPROTANT). There are lots of related work that have not been mentioned in this paper. For example, the following works were all focusing on cardiac segmentation.

Answer: The relevant part has been revised in the paper.

2. Please discuss the difference in detail between your method and the one in reference [12]

Answer: 1. We use the dice function as a loss function. 2. In addition to adding the ori[i-1] and label[i-1] branches, the ori[i+1] branch is added. 3. We used all the slices of each 3D image.

3. In your first contribution, you mentioned segmentation of a heart, which includes RV, LV and Myocardium. However, in experiments, only myocardium is segmented, why? Can you method segment all anatomies? If so please conduct experiments.

Answer: This method also has a good effect on the segmentation of LV and RV (we conducted experiments on the dataset of West China Medical College, Sichuan University). However, our ACDC dataset does not have split labels for LV and RV, so experiments cannot be performed.

4. Only one dataset (ACDC) was used. I would like to see experiments on different datasets, such as Sunnybrook cardiac data and 2011 LV segmentation challenge.

Answer: I did not successfully get these two data sets. However, I conducted an experiment on the heart data set provided by West China Hospital of Sichuan University. The results of this experiment have been reflected in the paper.

Reviewer 2:

The architecture described in this work is in some way similar to others approaches proposed in the literature exactly in the same context as this paper. For example, in [3] the authors propose a way to introduce contextual information into the NN. And in [1] the authors studied the effects of introducing the original information into the encode path of the u-net. Can the authors discuss the differences between those approaches and the one proposed?

Answer: Our approach combines the contextual sequence information of the image with the appropriate loss function and training strategy.

Did the authors consider using the Dice coefficient or Jaccard as the objective function, instead of the cross entropy ? And why?

Answer: I tried using the dice function as a loss function, but the effect is not as good as cross entropy, so I don't use the dice function.

Given that little training data is available, why are the authors not using data augmentation ?

Answer: Considering the fact that the heart data itself is relatively noisy and the contrast between the target area and surrounding tissue is small, no data enhancement method is used.

5 The paper should describe precisely how the dataset were selected for training, validation and testing.

5.a) Were the dataset split by patients or slices ?

5.b) How many images were used for validation (i.e. training the hyperparameters) ?

5.c) Was the validation dataset the same as the one used for testing or training ?

Answer: It has been explained clearly in the article.

6. Remove or translate the comment to english in Fig. 9

Answer：It has been modified

7. First sentence of the 4 paragraph (line 280) is not clear, it should be rewritten.

Answer：It has been modified

---

## [Decision Letter · Decision Letter 1]

28 Oct 2019

PONE-D-19-18130R1

Cardiac MR Segmentation Based on Sequence Propagation by Deep Learning

PLOS ONE

Dear Dr. Gao,

Thank you for submitting your manuscript to PLOS ONE. After careful consideration, we feel that it has merit but does not fully meet PLOS ONE’s publication criteria as it currently stands. Therefore, we invite you to submit a revised version of the manuscript that addresses the points raised during the review process.  In particular, the reviewers cite a number of additional results that should be added to the analysis, and also point out several corrections.  Furthermore, the manuscript is currently not in compliance with PLOS ONE's policy on data and software sharing.  It was noted by Reviewer 3, that some of the data used in the manuscript has not been made publicly available.  Please also clearly indicate where source code implementing the proposed algorithm has been made available.  For more information, see the Data Availability policy and the Materials and Software Sharing policy.

We would appreciate receiving your revised manuscript by Dec 12 2019 11:59PM. To enhance the reproducibility of your results, we recommend that if applicable you deposit your laboratory protocols in protocols.io, where a protocol can be assigned its own identifier (DOI) such that it can be cited independently in the future. For instructions see: http://journals.plos.org/plosone/s/submission-guidelines#loc-laboratory-protocols

We look forward to receiving your revised manuscript.

Kind regards,

Dzung Pham

Academic Editor

PLOS ONE

Reviewers' comments:

Reviewer's Responses to Questions

**Comments to the Author**

1. If the authors have adequately addressed your comments raised in a previous round of review and you feel that this manuscript is now acceptable for publication, you may indicate that here to bypass the “Comments to the Author” section, enter your conflict of interest statement in the “Confidential to Editor” section, and submit your "Accept" recommendation.

Reviewer #1: (No Response)

Reviewer #3: (No Response)

2. Is the manuscript technically sound, and do the data support the conclusions?

Reviewer #1: Partly

Reviewer #3: Partly

3. Has the statistical analysis been performed appropriately and rigorously? 

Reviewer #1: Yes

Reviewer #3: No

4. Have the authors made all data underlying the findings in their manuscript fully available?

Reviewer #1: Yes

Reviewer #3: No

5. Is the manuscript presented in an intelligible fashion and written in standard English?

Reviewer #1: Yes

Reviewer #3: No

6. Review Comments to the Author

Reviewer #1: I would like to see the segmentation performance of the proposed method on three anatomies, i.e. RV, LV and myocardium. At the moment, only myocardiumis shown. Please conduct more experiments.

Reviewer #3: Review summary:

The authors propose an encoder-decoder architecture to perform the automatic segmentation of the myocardium in cardiac MR 3D images. Their approach is based on the work of (Zheng et al, TMI 2018) which was dedicated to the integration of volume information in 2D segmentation convolutional neural networks. Though the proposed network contains very limited innovation on methodology compared to the original paper (addition of one branch containing the next image to segment), the improvement on the geometrical segmentation scores compared to the baseline is significant and supports the interest of the designed model for this particular task. It is therefore our belief that the study has merits and is of interest for the journal.

However, the paper fails to address several key requirements for publication in the PLOS journal:

- on the performed evaluation, which is incomplete and unfair to the state-of-the-art (I)

- on the conducted experiments (II)

- on the data, not fully available (III)

- on the drawn conclusions (IV)

We also noted:

- two misunderstandings on the theory of deep learning (V)

- several errors in the writing, with sentence repetitions and numerous grammar errors (VI)

- little effort in the response to the reviewers during the first round of modifications, on key aspects of the questions (VII)

---

The details of each aspect is presented here under.

I) Evaluation

The authors did a good job in performing cross-validation of several models on a complementary set of metrics and on two datasets. However, the authors do not directly compare to the results already published on the ACDC dataset (https://www.creatis.insa-lyon.fr/Challenge/acdc/). In particular :

- a different set of metrics is used (Dice, area under curve, F1 score, and JSC) instead of Dice and HD. The formulas are given for half of them (please give all or none). This choice of metrics is not argued, even though most of the metrics used are more relevant to evaluate classification than segmentation.

- from the description it appears the metrics were applied on 2D slices instead of the volumes, which is not suitable for a 3D problem, especially considering that the authors propose a 3D approach of the problem.

- when looking at results from the other teams in 2017, published in (https://www.creatis.insa-lyon.fr/Challenge/acdc/files/tmi_2018_bernard.pdf), it appears that the winner of the challenge obtained better results on the myocardium than those presented in the paper with regard to the only shared metric (average dice of 0.91 against 0.88 in the proposed paper). Also, it is untrue that no 3D method was proposed.

- only one structure is segmented, which is unfortunate considering that the ACDC dataset proposes annotations for three (left ventricle, myocardium, and right ventricle) and that the method is generic.

- only the average performance is given. We argue standard deviations are equally important to represent the robustness of a segmentation model.

- No scores are given for the second dataset (only visuals on three cases) which does not allow to have an idea on the generalization of the approach.

II) Missing experiments

Two experiments are greatly missing in order to support the authors’ claim:

- an ablation study showing the improvement of the added branch with ori[i+1] (Fig.1) is essential, if only to discuss the behavior of the 3D approach proposed in (Zheng et al., TMI 2018). As a new pre-processing step is used, its effect should also be discussed.

- local evaluation of the geometrical scores is required to prove the approach is better than classical 2D models (here U-Net) that segment slices independently. Indeed, 2D models are known to fail at the apex and the base of the heart, as discussed in the original ACDC paper. This local failure was identified as the main limitation. The addition of context in a 3D model as the one proposed here is in particular aimed at improving the accuracy in those regions. Therefore, statistics restricted to those regions should be presented.

III) The second dataset, from the West China Hospital of Sichuan University and consisting of 150 patients, is not linked to any publication or internet site, and cannot be found online from the name. It therefore appears that this dataset is novel. To meet the requirements of the journal, access to the dataset should be given at time of submission. Also, a more thorough introduction of the dataset properties, in variety, pathology, and annotation should be added to the paper.

IV) It is mentioned in the conclusion that the network “effectively combines the sequence features”, performs “quick” segmentation, is “more robust”, is “suitable for the segmentation tasks of other medical images”. The first claim lacks the ablation study mentioned in II) to stand. The second has not been evaluated in any way. The third is backed up by an interesting comparison of the average performance on the 5 experiments of the cross-validation. However, this should include standard deviations to be meaningful. Finally, the last claim is out of context and not backed by evidence.

V) The authors present two misconceptions on basic deep learning theory. Firstly, it is mentioned that the pooling layer (one has to look at Fig1 to know it is max pooling) “can loose useless features that do not utilize segmentation, effectively reducing the complexity of the network, reducing the number of parameters, making the network train faster and occupying less computing resources”. This description is not accurate and partly wrong. Pooling layers reduce feature maps spatially, which improves the resilience to linear transformations, and indeed limits the needs in memory and probably speeds up the training phase. However, the number of parameters and the complexity is unchanged (it is only linked to the number and size of features and the number of input feature maps). Please review your interpretation of the pooling effect. Secondly, it is said that “if the learning rate is too high, it will lead to over-fitting”, which is false. A high learning rate can cause a network to have a noisy convergence, and even diverge. A small learning rate and a sufficient capacity and number of epochs is needed for a network to over-fit. That is why a validation set is often used to select the right model parameters as the one performing best on the validation set. The authors are encouraged to look into the interpretation of learning curves and learning rates.

VI) We here report the main changes that should be done in the writing.

- line 38, please do not refer to authors as “Paper”, but state the full name of the team and date of publication, such as (first author name et al., date).

- line 61, wrong sentence continuity. We propose “method, effectively overcoming”

- line 83, as it is not common knowledge, please rephrase your sentence. For instance “ No deep learning method has been used for cardiac MRI prior to 2013” + give citation of the paper in 2013 you are thinking of.

- line 100, wrong sentence continuity “ convolutional networks that provided good results on the ACDC 2017”

- line 118, “ori” instead of “orion”

- line 120 “deep supervision” instead of “depth”

- line 123, the sentence is unclear, especially when writing “designing a convolution of the number of different convolution kernels”. Please make your message clearer.

- line 126, what is the fitting rate of a network? Please use standard names.

- line 128 to 130, this sentence appears twice.

- line 143, a sentence should not begin by “And”

- line 153, “Cross-Entropy is the most commonly employed”

- line 167, “the myocardial annotations”

- line 217, “validation” instead of “test”, otherwise, if the selection of the model is indeed performed on the test set, the experiments are skewed and should be redone without involving the test set in the training phase.

- line 245 to 260, avoid repetitions. For instance, use “ Figure [] shows “ instead of “As shown in Figure[], this figure is”

- line 265, “our method has much better”

- line 269 to 271, please be careful on the punctuation and the splitting of ideas in sentences.

-line 278 to 283, this section should be mentioned in the dataset description. Information should be given on the splitting (how many patients for train, test, valid etc.), and scores should be added.

VII) We find the authors did not make enough efforts to take into account the remarks of the reviewers. In particular, the needs to perform evaluation on all structures and with ablations of the innovations were already mentioned respectively by reviewer 1 and reviewer 2, but not taken into account. In particular, it was answered that the proposed model does work better on other structures but that the ACDC dataset only had myocardium annotations, which is incorrect.

---

Changes required:

All points described above should be addressed for the paper to be acceptable. This involves in particular to:

- give the results on all the annotated structures of the ACDC dataset, with standard deviations in addition to mean values.

- give results for the experiments on the second dataset, and a thorough description of it.

- perform ablation studies and local evaluation to show and discuss the interest of the method compared to a classic U-Net and to the network in (Zheng et al, TMI 2018). If necessary, the results of the other models could be removed, as they are not related to this comparison but instead reinforce the choice of a U-Net like architecture (they all perform worse than U-Net).

- give in a table or a figure the results of the proposed method restricted to local sections of the heart (apex, base, middle) in order to discuss the whereabouts of the improvement. If necessary, this could replace the figures from 4 to 6 which support the same message. Besides, these figures could fit in one to save a lot of room.

- Refine the writing and avoiding misleading interpretations of hyper-parameters.

7. PLOS authors have the option to publish the peer review history of their article (what does this mean?). If published, this will include your full peer review and any attached files.

Reviewer #1: No

Reviewer #3: No

---

## [Decision Letter · Decision Letter 2]

2 Mar 2020

Cardiac MR Segmentation Based on Sequence Propagation by Deep Learning

PONE-D-19-18130R2

Dear Dr. Gao,

We are pleased to inform you that your manuscript has been judged scientifically suitable for publication and will be formally accepted for publication once it complies with all outstanding technical requirements.

With kind regards,

Dzung Pham

Academic Editor

PLOS ONE

Additional Editor Comments (optional):

Reviewers' comments:

Reviewer's Responses to Questions

**Comments to the Author**

1. If the authors have adequately addressed your comments raised in a previous round of review and you feel that this manuscript is now acceptable for publication, you may indicate that here to bypass the “Comments to the Author” section, enter your conflict of interest statement in the “Confidential to Editor” section, and submit your "Accept" recommendation.

Reviewer #3: (No Response)

2. Is the manuscript technically sound, and do the data support the conclusions?

Reviewer #3: Yes

3. Has the statistical analysis been performed appropriately and rigorously? 

Reviewer #3: Yes

4. Have the authors made all data underlying the findings in their manuscript fully available?

Reviewer #3: Yes

5. Is the manuscript presented in an intelligible fashion and written in standard English?

Reviewer #3: Yes

6. Review Comments to the Author

Reviewer #3: (No Response)

7. PLOS authors have the option to publish the peer review history of their article (what does this mean?). If published, this will include your full peer review and any attached files.

Reviewer #3: No

---

## [Editor Report · Acceptance letter]

9 Mar 2020

PONE-D-19-18130R2 

Cardiac MR Segmentation Based on Sequence Propagation by Deep Learning 

Dear Dr. Gao:

I am pleased to inform you that your manuscript has been deemed suitable for publication in PLOS ONE. Congratulations! Your manuscript is now with our production department. 

With kind regards,

on behalf of

Dr Dzung Pham 

Academic Editor

PLOS ONE